



# Aerodynamic Performance of the NREL S826 Airfoil in Icing Conditions

Julie Krøgenes[1], Lovisa Brandrud[1], Richard Hann[2], Jan Bartl[1], Tania Bracchi[1], and Lars Sætran[1]

[1]Department of Energy and Process Engineering, Norwegian University of Science and Technology
[2]Department of Engineering Cybernetics, Norwegian University of Science and Technology

*Correspondence to:* Julie Krøgenes (juliekrogenes@gmail.com)

**Abstract.** The demand for wind power is rapidly increasing, creating opportunities for wind farm installations in more challenging climates. Cold climate areas, where ice accretion can be an issue, are often sparsely populated and have high wind energy potential. Icing may lead to severely reduced aerodynamic performance and thereby reduced power output. To reach a greater understanding of how icing affects the aerodynamics of a wind turbine blade, three representative icing cases; rime ice, glaze ice

and a mixed ice, were defined and investigated experimentally and computationally. Experiments at Re$= 1.0 \times 10^5 - 4.0 \times 10^5$ were conducted in the low-speed wind tunnel at NTNU on a two dimensional wing with applied 3D-printed ice shapes, determining lift, drag and surface pressure distributions. Computational results, obtained from the Reynolds Averaged Navier-Stokes fluid dynamics code FENSAP, complement the experiments. Measured and predicted data show a reduction in lift for all icing cases. Most severe is the mixed ice case, with a lift reduction of up to $30\%$ in the linear lift area, compared to a clean refer-

ence airfoil. Computational results show an under-prediction in maximum lift of $7 - 18\%$ compared to experimental values. Curvature and tendencies for both lift and drag show good agreement between simulations and experiment.

## 1 Introduction

### 1.1 Background

Wind power is one of the world's fastest growing sources of electricity production, with the global installed capacity having

increased from 24 GW in 2001 to 500 GW in 2016 (GWEC, 2016). Several factors indicate continued growth in the wind power industry. The Paris agreement from 2008 contributed to increased global initiative towards renewable energy, and decreasing wind prices make wind energy production competitive in new markets. This growth comes with the need for wind farm installations in new areas. Cold climate and high altitude areas have high wind energy potential, due to the dense, cold air and high wind speeds (Fortin et al., 2005). In addition, cold climate areas are often sparsely populated and therefore less

sensitive to visual- and noise pollution. However, wind farms located at high latitudes or close to mountains can be exposed to frequent icing events in cold periods (Tammelin and Säntti, 1998).

For wind turbine operation, there are several problems related to icing. Increased risk of structural fatigue, measurement errors in ice covered equipment, safety hazards of ice throw, electrical and mechanical failures are some examples (Parent and



Ilinca, 2010). Icing can also lead to overproduction and thus excessive structural loads, due to sudden increase in momentum, which the wind turbine is not dimensioned for (Jasinski et al., 1998).

As a result of accumulated ice on a turbine blade surface, the airfoil geometry changes. This will have an impact on aerodynamic properties and thereby wind turbine power output. It has been shown that even the slightest ice accretions can lead to a 20% power output reduction (Bose and Rong, 2009). Consequently, there are often large deviations between predicted and actual power curves of wind turbines in areas vulnerable to icing (Barber et al., 2009b). More knowledge about these deviations is important in order to determine the expected energy production of a project, and thereby its economic viability.

With applied icing protection systems (IPS), negative effects of icing can be mitigated. Several systems are being developed or are already in use. When choosing the appropriate IPS, it is important to consider the ratio between recovered power output and power consumption of the chosen IPS. The optimal mitigation measure varies with the type and amount of ice (Parent and Ilinca, 2010).

Icing effects in the aircraft industry have been studied extensively both computationally and experimentally (Addy, 2000), (Bragg et al., 2005). Some of this research is applicable also to the wind power industry. However, due to different airfoil geometries and operation at lower Reynolds numbers, more research is needed in this area. Further development in computational fluid dynamics, through experimental validation, will make information more available and less expensive to obtain when evaluating new challenges.

## 1.2 Objective

The current study aims to obtain more knowledge about the effects of different ice accretions. This may, in turn, help wind farm developers in quantifying the realistic production potential of a specific location, in addition to choosing optimal ice protection solutions.

Another important objective is the validation of computational methods and their ability to predict flow around complex airfoil geometries. Therefore a combined numerical and experimental study was conducted. Three types of ice accretions were defined based on typical conditions for cold climate areas suitable for wind power production. Their effects on aerodynamic properties were investigated.

## 2 Method

This study investigates the aerodynamic performance of the NREL S826 airfoil, with 3D-printed ice shapes attached. The airfoil was designed by Somers (Somers, 2005) at the National Renewable Energy Laboratory and is intended for use at the blade tip of $20 - 40\,m$ diameter horizontal axis wind turbines. The blade tip is the part that is most exposed to icing due to large tip velocities leading to high accumulation rates. A 2D model of the S826 profile constructed of CNC-milled stiff Ebazell foam, with a chord of $0.45\,m$, a span of $1.18\,m$ and a hydraulically smooth surface was used for the experiments (Aksnes, 2015).



## 2.1 Defining Icing Cases

For the generation of 2D ice shapes, the LEWICE code (version 3.2.2) was applied (Wright, 2008). LEWICE is a widely used 2D ice accretion tool that has been validated over a large range of parameters (Wright, 1999). LEWICE has not been validated specifically for the low-Reynolds regime as it has been developed mainly for aircraft purposes. However, the numerical methods

implemented in the code are not excluding low-Reynolds numbers in particular and it is therefor assumed that the results are accurate enough for this study.

Icing cases are generally defined by the following parameters: free-stream velocity $V_\infty$, duration of icing $t_{icing}$, airfoil chord length $c$, angle of attack $AoA$, liquid water content $LWC$, median volume diameter $MVD$ and ambient temperature $T_\infty$. For this study, a large number of combinations of these parameters have been evaluated in order to find representative ice shapes

to investigate in detail. Three ice shapes were selected, which are mainly distinguished by the temperature at which they form. The selected cases are summarized in Table 1 and the resulting ice shapes are seen in Fig. 1. The liquid water content was adjusted according to empirical correlations of droplet size and water content for stratus clouds (CFR 14, 2006). It should be noted that the selected ice shapes may not be entirely representative for each icing type, as ice shapes vary extensively over the parameters stated above.

At very low temperatures, all droplets freeze on impact and form rime ice. Due to entrapped air between the frozen droplets, rime appears as white and displays rugged, rough surface. Glaze ice is an ice type that forms at temperatures close to freezing conditions. It is dominated by a low mass fraction of particles that freeze on impact. The majority of droplets form a liquid water film on the surface of the airfoil, which will either freeze or evaporate. Aerodynamic friction causes the liquid film to flow downstream as so-called runback. Glaze typically appears as transparent ice with a smooth surface. Mixed icing is an ice type

that is formed in the temperature regime between rime and glaze. Therefore, it is characterized by a balanced ratio between instantaneous freezing and surface freezing (Kraj and Bibeau, 2009). Due to this characteristic, the mixed ice builds up ice horns at an approximately $45°$ angle. In order to obtain a more extreme ice shape, the flow velocity was increased compared to the other cases, which leads to a larger accretion of mass.

**Table 1.** Parameters used to define ice shapes

| Parameter | Rime | Glaze | Mixed |
|---|---|---|---|
| $V_\infty$ | $25\,m/s$ | $25\,m/s$ | $40\,m/s$ |
| $T$ | $-10°C$ | $-2°C$ | $-4°C$ |
| $LWC$ | $0.43\,g/m^3$ | $0.34\,g/m^3$ | $0.55\,g/m^3$ |
| $MVD$ | $20\,\mu m$ | $30\,\mu m$ | $20\,\mu m$ |
| $t_{icing}$ | $40min$ | $40min$ | $40min$ |
| $AoA$ | $1°$ | $1°$ | $1°$ |
| $c$ | $0.3\,m$ | $0.3\,m$ | $0.3\,m$ |





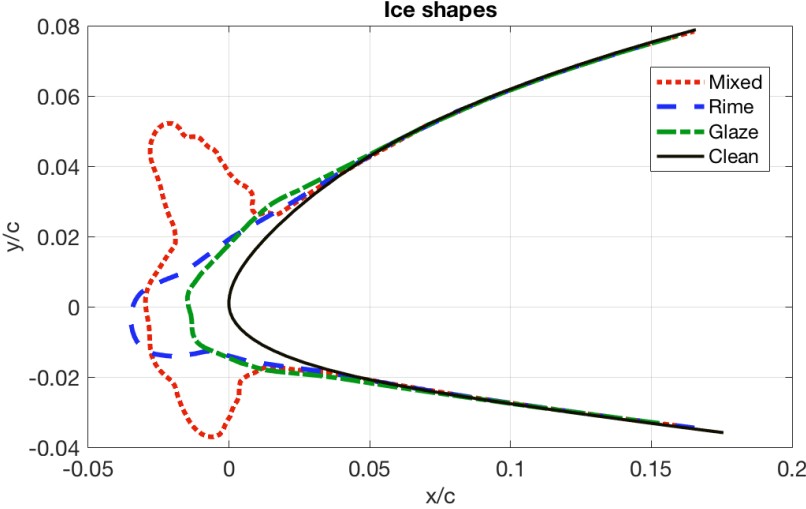

**Figure 1.** Modelled ice shapes on the leading-edge of the NREL S826 airfoil

The surface roughness $k_s$ for each icing case, seen in Table 2, was approximated by using empirical correlations (Shin and Bond, 1992). Generally, surface roughness development is driven by temperature and velocity, but also by droplet size. In cases with a significant amount of instantaneous freezing (rime and mixed), the roughness will be larger than for cases with surface freezing (glaze).

5     Models of the icing were created in the computer-aided design tool NX (NX, 2016). The roughness of the ice shapes was modelled as half spheres in a hexagonal packing arrangement. Physical ice models were 3D-printed in a PLA plastic, and the accuracy of the roughness was determined to be $\pm 0.05\,mm$.

## 2.2    Experimental Setup

The experimental work was carried out in the closed-loop low-speed wind tunnel at the Norwegian University of Science 10   and Technology (NTNU). The dimensions of the test section inlet are $1.8\,m \times 2.7\,m \times 12.0\,m$ (height$\times$width$\times$length), with the height increasing to $1.85\,m$ at the end of the test section to compensate for wall boundary layer growth. Measurements were conducted for AoAs ranging from $-8°$ to $18°$ degrees and for Reynolds numbers (Re) $1 \times 10^{-5}$, $2 \times 10^{-5}$, $3 \times 10^{-5}$ and $4 \times 10^{-5}$, with inflow turbulence intensities of $0.71\%$, $0.44\%$, $0.33\%$ and $0.31\%$, respectively. At AoA $= 18°$ the blockage ratio

**Table 2.** Surface roughness

| | |
|------|--------|
| Rime | $1\,mm$ |
| Glaze | $0.6\,mm$ |
| Mixed | $1\,mm$ |





$(A_{wing}/A_{tunnel})$ is 5.1% which is below the limit of 7.5% where blockage correction is considered to be required (Barlow et al., 1999). Additionally, earlier experiments with the same wing and wind tunnel show negligible blockage effects for this range of angles (Aksnes, 2015). Thus, the results in this paper are presented without blockage corrections.

Fig. 2 shows the experimental setup. In order for the experiments to resemble 2D flow over the wing span, two 0.3 m wing
5  elements were placed above and below the model wing. The ice shapes were attached to the leading edge of the main section and the dummies using insulating tape.

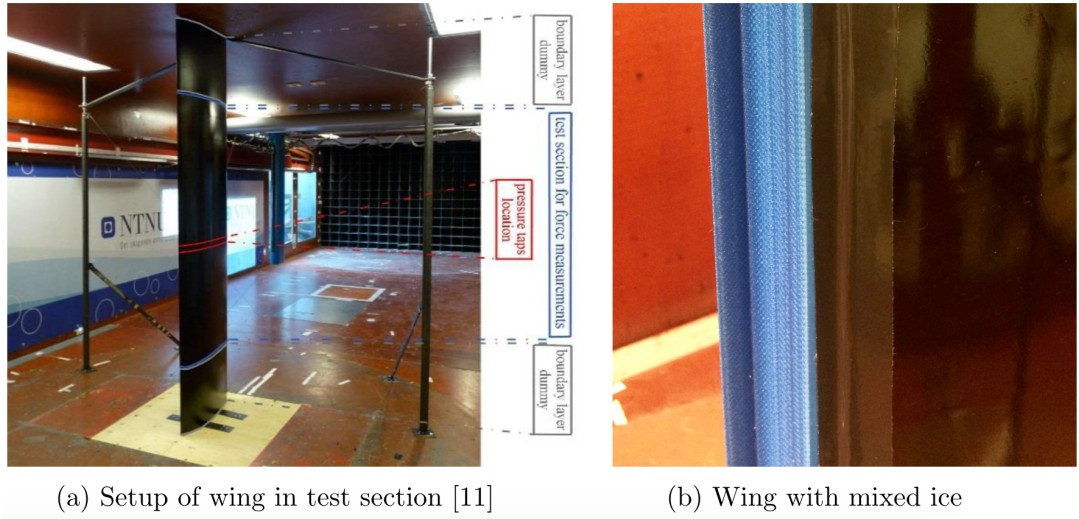

(a) Setup of wing in test section [11]  (b) Wing with mixed ice

**Figure 2.** Experimental setup

### 2.3   Measurement Methods

To determine the lift and drag of the wing, force- and pressure measurements were applied.

*Force Balance*

10  A force balance with three acting load cells, two in the flow direction and one perpendicular to the flow, was used to determine the lift force acting on the airfoil. The wing was mounted on the force balance, with no contact with other parts of the setup. The force balance was rotated to change the angle of attack, with a measured accuracy of $\pm 0.25°$. Measurements were performed in $60 s$ intervals with a sampling rate of $1000 Hz$. Normalized lift $C_L$ was derived by

$$C_L = \frac{F_L}{\frac{1}{2}\rho U_\infty^2 A} \tag{1}$$





where $F_L$ is the measured lift force, A is the area of the wing and $U_\infty$ is the freestream velocity. For all test cases, including with icing, the area was calculated using the clean wing chord length.

*Surface Pressure*

Surface pressure measurements were conducted to determine the pressure distribution around the airfoil. 32 pressure taps at the surface of the midsection were connected with plastic tubes to a pressure scanner consisting of an array of silicon piezoresistive pressure sensors, mounted inside the wing. Gage pressure was measured directly and static pressure upstream was used as reference pressure. Data was sampled for $60 s$ at a sampling rate of $333\,Hz$. Normalized pressure, $C_p$, was derived by

$$C_p = \frac{p}{\frac{1}{2}\rho U_\infty^2} \tag{2}$$

where $p$ is the measured pressure. Because of the attached ice, covering some pressure taps on the leading edge, obtained surface pressure distributions exclude the measurements from these taps. This means up to $c = 0.02$ on both pressure- and suction side of the rime- and mixed ice. The glaze ice was covered up to $c = 0.05$ on the pressure side and $c = 0.03$ on the suction side.

*Wake Rake Measurements*

To determine pressure- and skin friction drag, wake rake surveys were conducted. The wake rake consists of 21 uniformly distributed tubes of 1mm diameter, with 10 mm spacing between the center of each tube. It was placed $0.7c$ downstream of the trailing edge, at the same height as the midsection of the wing. The tubes, as well as the stagnation pressure from a pitot tube placed 5c upstream, were connected to the same pressure scanner as described above. Pressure measurements were performed at a sampling rate of 100 Hz for a duration of 30 seconds at each angle of attack. Drag is calculated by integration of momentum deficit found by measuring the axial velocity profile in the wake of the airfoil. By applying the 2D continuity and momentum balance to a control volume around the airfoil, as proposed by Chivaee (Chivaee, 2014), the drag coefficient $C_D$ can be defined as

$$C_D = 2 \int \frac{u}{U_\infty} \left(1 - \frac{u}{U_\infty}\right) d\left(\frac{y}{c}\right) \tag{3}$$

where $u$ is the velocity in the wake, $U_\infty$ is the freestream velocity and y is the width of the wake. From the Bernoulli equation, the velocities can be computed from the measured pressure

$$H = p_0 + \frac{1}{2}\rho u^2 \tag{4} \qquad\qquad u = \sqrt{\frac{H - p_0}{\frac{1}{2}\rho}} \tag{5}$$



where $H$ is the stagnation pressure in the wake and $p_0$ is the static reference pressure at the control volume surface. The position $0.7c$ was considered a sufficient distance downstream for the static pressure in the wake to have stabilized. The static pressure through the test section was assumed to be constant.

Lastly, it is well known that wake rake surveys are only valid for angles of attack before stall and where separation is not
present, as it cannot capture a 3D velocity field. This means up to around $12°$ for the clean airfoil, and less for the icing cases depending on the shape.

### 2.4 Measurement Uncertainties

Statistical uncertainties in the calculated lift- and drag coefficients were estimated following the method proposed by Wheeler and Ganji (Wheeler and Ganji, 2004). Systematic errors were expected to be the largest contributor, as opposed to precision
errors, and were therefore the main focus.

Taking into account systematic errors in velocity- and load cell calibration, the expected error in lift coefficients was found to be $\pm1.4\%$ for all AoA.

With regards to the drag calculations, the main error considered was the influence of variations in static pressure. Wind tunnel investigations showed minor losses in static pressure from the pitot probe upstream to the position of the wake. Additionally,
measurements indicated a slightly non-stabilized static pressure at 0.7c downstream, resulting in a decrement in the wake static pressure. Both mentioned effects contribute to a reduction in the calculated drag. Uncertainty estimations show an offset of approximately $\Delta C_D = 0.01$ in the calculated drag, over the applied range of AoA, due to static pressure effects.

### 2.5 Simulation Setup

The steady-state flow field around the iced geometries was solved with FENSAP, a state-of-the-art Navier-Stokes CFD solver
(Habashi et al., 2004). The solver is part of the software package FENSAP-ICE which is a 3D icing simulation tool. In this study, for the sake of simplicity, LEWICE was used for the ice generation and FENSAP only as a flow field solver.

For low Reynolds numbers with free transition, CFD is typically unable to predict aerodynamic characteristics accurately. The occurrence of laminar separation effects are difficult to fully capture with common CFD methods. This is assumed only to be an issue regarding the clean airfoil, as the occurrence of ice provides surface roughness heights sufficiently large to trigger
laminar-turbulent transition at the leading-edge. Therefore, the calculations were performed fully turbulent. The turbulence model chosen was Spalart-Allmaras (Spalart and Allmaras, 1992), as it performs reasonably well for turbulent flows with negative pressure gradients.

The FENSAP calculations were run as 2D simulations with settings specified in Table 3. The simulations have been checked for transient separation behavior for all icing cases without any findings. Also, the experimental data showed no evident
transient effects. This justifies the assumption of steady-state calculations. The clean airfoil discretization has been executed as a structured O-grid with a full resolution of the boundary layer and a total of approximately 80 000 cells. Meshing iced



geometries is challenging due to the occurrence of large convex and concave curvatures. Experience shows that the best results are achieved by using hybrid O-meshes. The iced geometries were discretized with a structured boundary layer and an unstructured far field. In order to deal with the complex curvatures and to limit the required computational capacities, the meshes contained between 40 000-50 000 cells. Several grid sizes have been tested in order to ensure that the results are

independent from the grid.

**Table 3.** Computational Settings

| | |
| --- | --- |
| Momentum equation | Navier-Stokes |
| Energy equation | Full PDE (partial differential equations) |
| Turbulence model | Spalart-Allmaras |
| Time step | Steady-State |
| CFL number | 100 |
| Artificial viscosity | Streamline upwind |
| Convergence criteria | 1e-8 |

## 3 Results and Discussion

This section presents comparisons of lift, drag and surface pressure distributions for the different ice shapes, found from wind tunnel measurements, in addition to computational results for the same cases. Reynolds numbers close to $1 \times 10^6$ are typical for NREL S826 operation. The experiments where conducted at lower Reynolds numbers ($1 \times 10^5$ - $4 \times 10^5$) due to wind tunnel

limitations, hence the transferability of the results to higher Re is limited. However, the objective of this study is to discover general trends of icing impact on aerodynamic performance, so the Re mismatch is considered to be not significant. The main results are shown for the highest investigated Re $= 4 \times 10^5$ as it is closest to the design point and is least affected by low-Re effects.

### 3.1 Effects of Icing on Airfoil Coefficients

Experimental lift results, at Re $= 4 \times 10^5$, are shown in Fig. 3. For all three ice cases, lift is decreased relative to the clean airfoil. Rime and glaze curves follow each other closely, reducing lift with approximately $10-15\%$ in the linear lift region. For glaze ice, at all Re, the linear lift coefficient incline is interrupted by a lift drop at about AoA $= 10°$ followed by a steep increase at about AoA $= 13°$. This behavior is not seen in the simulations, and the reason for this is not clear at this point.

The mixed ice lift curve differs from the other ice cases. In the linear area, mixed ice lift is $25-30\%$ lower than the clean

case lift and $15-20\%$ lower than the rime ice lift. Stall is reached at AoA $= 7°$, whereas the clean case, rime- and glaze ice stall at $12-13°$.

It was concluded by Homola et. al (Homola et al., 2011) that a $17\%$ lift reduction due to leading edge rime ice accretion, caused a $28\%$ power curve reduction. The results are supported by Barber et al. (Barber et al., 2009a), Seifert and Richard



(Seifert and Richard, 1998) and Laakso and Peltola (Laakso and Peltola, 2005) who all found 20-30% power curve reduction from resembling ice accretions. The current study shows that mixed ice causes more severe performance losses than rime- and glaze ice, leading to the assumption that less streamlined ice shapes can reduce power output even more.

Drag coefficient curves, presented in Fig. 3, clearly show that icing leads to increased drag compared to the clean airfoil.

Rime- and glaze ice show similar tendencies, while mixed ice initiates a more extreme performance degradation. At AoA =7°, which is within the normal operating range of AoA for the NREL S826 (Somers, 2005), rime ice leads to 50% drag increase. Mixed ice is, at this angle, already in the stalled region, resulting in unstable flow behavior and 600% drag increase. As previously discussed, after stall, wake rake measurements have a considerably higher uncertainty. Still, these numbers give an indication of the impact certain ice types can have on aerodynamic performance. Because of extensive separation effects at

high AoA, drag could not be calculated at these angles for all cases, hence some drag curves end at lower AoA.

The NREL S826 was designed with the aim of achieving relatively constant drag from $C_L = 0.4$ to 1.2. Seen from lift coefficient curves for the clean airfoil, this interval ranges from AoA $= -2°$ to 6°. In this area, the clean airfoil drag is approximately constant. Rime- and glaze ice curves start to incline at an earlier stage, and the mixed ice curve even more so. This means that icing is negatively influencing the intended behavior of the airfoil and limiting the optimal operational

envelope.

The lift decrease and drag increase that can be observed for all ice shapes indicate that icing generally leads to reduced performance and hence power output losses. This will, as shown for the different test cases, depend on the icing type and resulting ice accretion. Rime- and glaze ice shapes are more streamlined, functioning like an elongation of the airfoil geometry. They show quite similar trends compared to the clean case, which is expected due to their shapes' resemblance. The mixed

ice case shows significantly lower lift and higher drag, clearly being the most severe ice accretion with regards to performance losses.

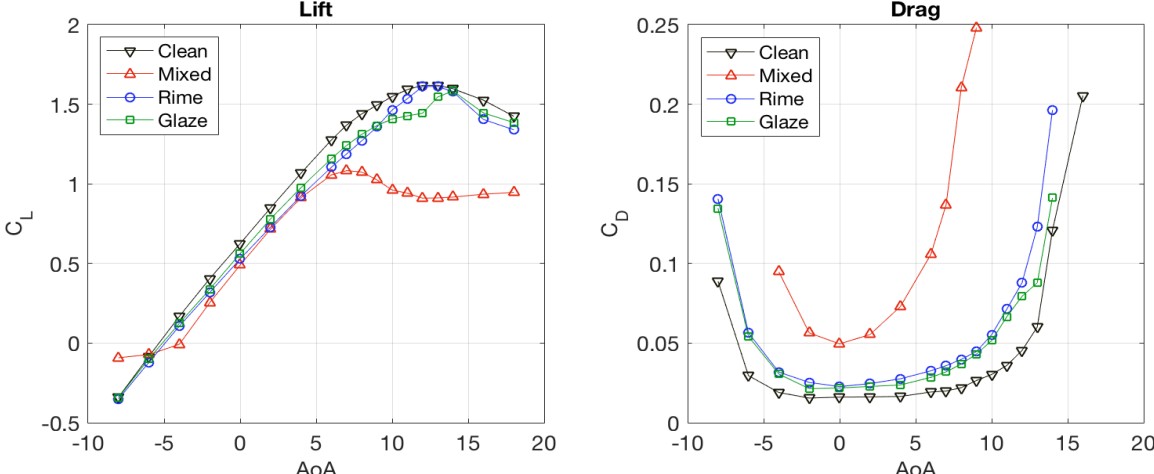

**Figure 3.** $C_L$- and $C_D$ curves, obtained from force balance- and wake rake measurements, respectively, at Re$= 4 \times 10^5$



## 3.2 Prediction of Airfoil Coefficients by FENSAP-ICE

As mentioned in the introduction, an aim of the present study was to validate the application of numerical tools to determine aerodynamic behaviour in the event of icing.

Computational lift values show good agreement with experimental values in the linear area, see Fig. 4, while it deviates
more around stall occurrence. Simulations show an under-prediction of lift in the stalled region. The curves, however, display a large resemblance.

The computational drag values follow the slope of the experimental curves for all the icing cases before stall occurs, as can be seen in Fig. 4. For post-stall AoAs, experimental curves are steeper than the computational curves. Deviations in this area is expected, due to the uncertainties mentioned for drag measurements at high AoAs. From the comparison, it can be
concluded that the computational results give a reasonable and useful estimation of the aerodynamic characteristics in the normal operating range of AoA.

The deviations seen in lift and drag can have several explanations. The Spalart-Allmaras turbulence model chosen for the simulation set up, assumes fully turbulent flow. For the complex shapes studied, at AoAs where separation occurs, the aerodynamic characteristics are affected in ways that are not necessarily captured by a simplified turbulence model. Additionally, the
simulations are conducted assuming 2D flow. The experiments were intended to represent 2D flow, however, some 3D effects have been observed that are not accounted for in this study (Bartl et al., 2017), (Prytz et al., 2016).

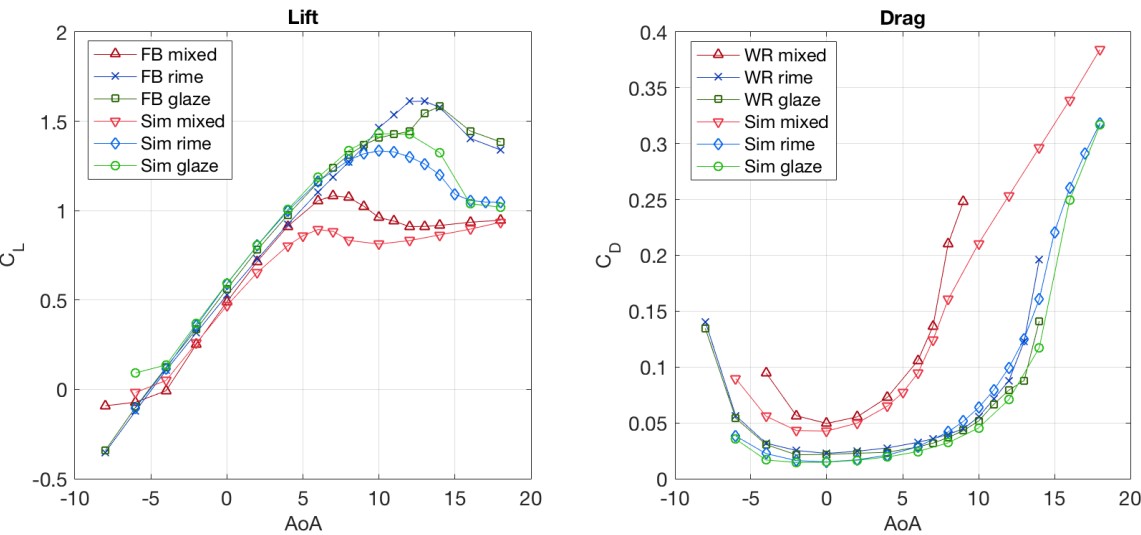

**Figure 4.** Force balance (FB) lift results and wake rake (WR) drag results for the icing cases compared to FENSAP simulations (Sim), at Re$= 4 \times 10^5$




### 3.3 Surface Pressure Distributions

Surface pressure results for the NREL S826 model airfoil, with and without icing, was investigated to obtain information about the effects of icing on pressure coefficients and local separation.

Fig. 5 shows the measured surface pressure distributions for the clean airfoil and the three icing cases at different AoAs. The cases with ice accretion show a smaller pressure difference between suction- and pressure side relative to the clean airfoil, which agrees with the previously discussed lift reduction due to icing. The clean airfoil surface pressure distribution, at AoA $8°$ and $12°$, shows a suction peak immediately downstream of the leading edge. This implies that for high AoA the majority of the total lift comes from the pressure difference within the first $50\%$ of the chord, making lift vulnerable to disturbances in this area. Leading edge icing causes an earlier onset of trailing edge separation, contributing to a reduction of maximum lift and stall angle.

The mixed ice curves at AoA $= 0°$ and $4°$ show low $C_P$ (high -$C_P$) over the ice shape. This may be an effect observable in combination with the formation of a leading edge separation, which is indicated by the relatively constant $C_P$ in this space (Lee and Bragg, 1999). Reattachment may be occurring when the perturbed flow (with ice) rapidly approaches the unperturbed flow (clean), as for mixed ice at AoA $= 0°$, $x/c = 0.15$. From AoA $= 8°$, reattachment seems no longer apparent. The leading edge separation and the trailing edge separation appear to connect fully over the entire surface. This corresponds well with findings presented earlier, showing mixed ice stall occurrence from around this angle. As the pressure holes underneath the ice models were covered, surface pressure information from this area was not available.

Rime- and glaze ice show no clear evidence of a separation bubble in the measured region. Mixed ice has a shape that extends both upwards and downwards, causing flow detachment on both the pressure- and the suction side. This flow detachment is likely to be the main cause for the decrease in maximum lift and the increase in drag. As a consequence the airfoil stalls at a lower angle of attack of AoA $= 5 - 6°$.





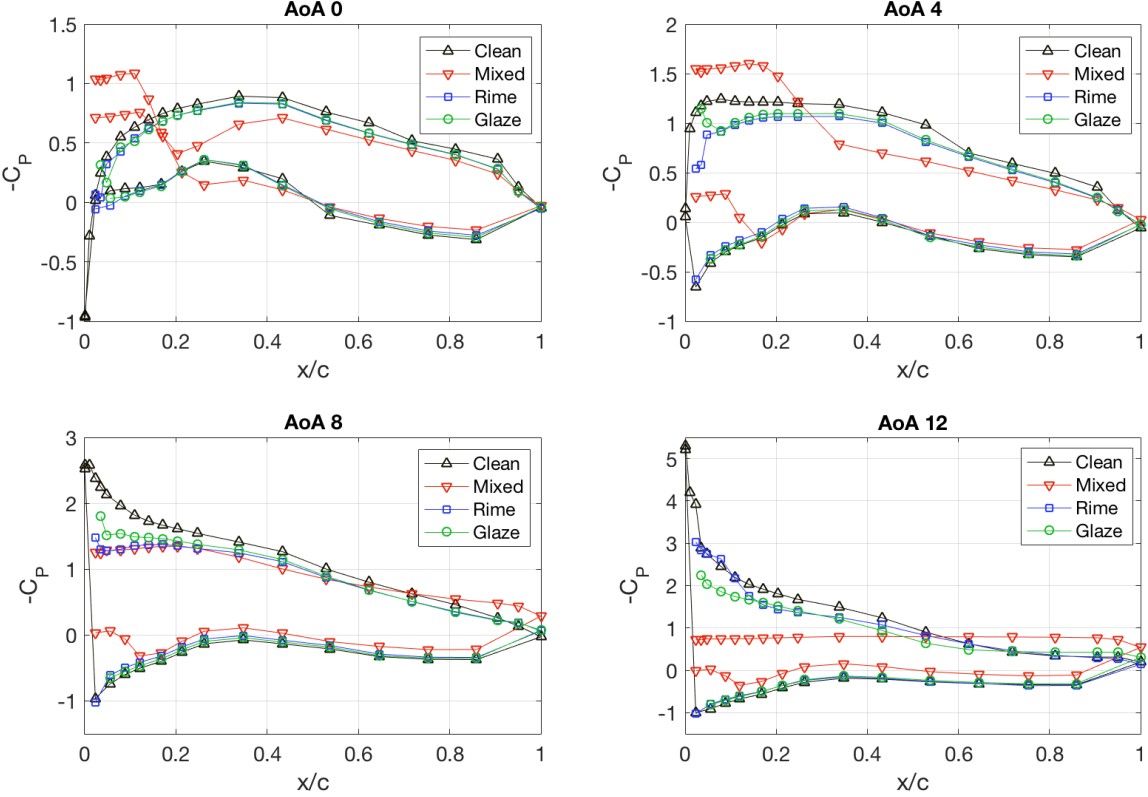

**Figure 5.** Experimental surface pressure distribution at $Re = 4 \times 10^5$

Fig. 6 shows a comparison between experimental- and computational $C_P$-curves. At AoA = $4°$, $x/c = 0.4$, both clean- and mixed ice simulations show a 10% lower $C_P$ on the suction side than experimental results. This difference increases to 14% and 35%, respectively, when increasing AoA to $8°$. Lift and drag results, discussed in the previous section, also show an increasing deviation between simulated and experimental values with AoA. Simulation under-prediction increases on the suction side, but

5   not significantly on the pressure side, with AoA. For low AoA, under-prediction on both sides evens each other out, resulting in little difference in $\Delta C_P$ and thereby lift, compared with experimental values. For higher AoA, $C_P$ under-prediction on the suction side is larger than on the pressure side, leading to a lower computed lift. One reason for this is likely the earlier onset of trailing edge separation on the suction side when icing is present, making it difficult to predict pressure correctly by the turbulence model.

10    All three icing shapes are likely to trigger laminar transition at the leading edge, and results show coherent behaviour. It is therefore assumed that we have a fully turbulent flow regime for these cases. For the clean airfoil, experimental results show indications for transition along the airfoil surface and for laminar separation bubbles. In order to investigate this further, XFOIL





(XFOIL, 2013) simulations were run with the same specifications and plotted against SP results, as presented in Fig. 7. From the XFOIL results laminar separation bubbles are more evident, occurring approximately at x/c = 0.6 on the suction side and at x/c = 0.45 on the pressure side for AoA = 0°. For AoA = 8° this can be seen at approximately x/c = 0.55 on both sides. A study by Bartl et al. (Bartl et al., 2017) show corresponding results for the same airfoil at low Re numbers. Since all FENSAP simulations are run with a fully turbulent flow regime, it is not able to capture laminar transitions, which can contribute to the deviations from experimental results for the clean airfoil.

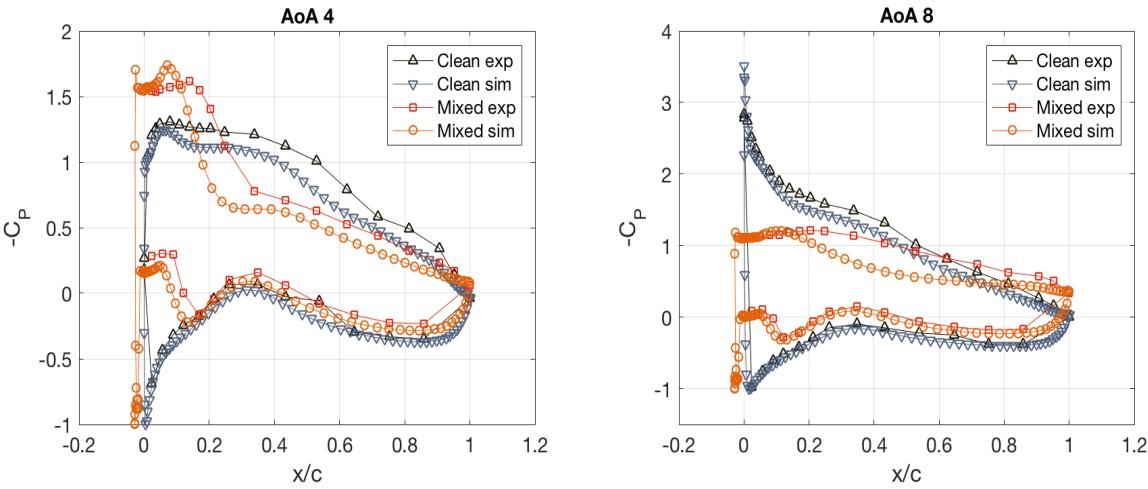

**Figure 6.** Surface pressure distribution for clean airfoil and mixed ice compared with FENSAP simulations at $Re = 2 \times 10^5$

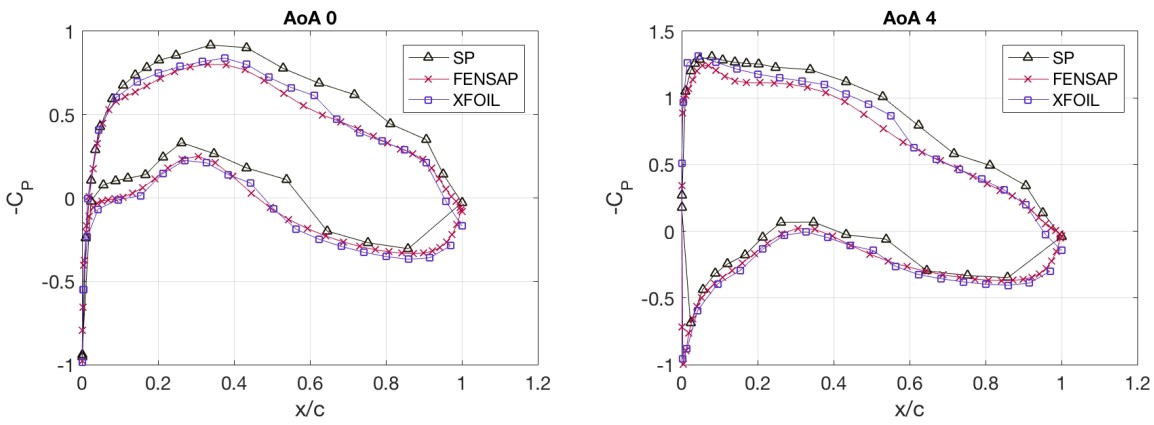

**Figure 7.** Clean airfoil surface pressure distributions obtained from surface pressure measurements (SP) FENSAP and XFOIL, at $Re = 2 \times 10^5$



### 3.4 Reynolds Number Dependency

The NREL S826 airfoil is intended for operation in Re $= 1 \times 10^6 - 3 \times 10^6$ (Somers, 2005). The effects of icing are studied here at lower Reynolds numbers, limiting the transferability of the results to the design Re. However, it is still relevant to investigate the influence of the Reynolds number on the effect of icing on the aerodynamics.

Ice accretion show similar effects on lift and drag for all Re investigated in this study, as shown in Fig. 8. Clean airfoil drag measurements for Re $= 1 \times 10^5$ deviate from the other curves. However, for Re numbers this low, there can exist large-scale vortices in the wake, similar to behind a cylinder, which will lead to rotational losses that are not detected by wake rake measurements (Mueller et al, 1983). This could likely be a reason for the deviations at low Re numbers. In addition, low flow velocities experience higher relative disturbances, adding uncertainty to the measurements. In summary, it seems that for the

investigated Re range, there is only a limited influence of the Reynolds number on the icing effects.

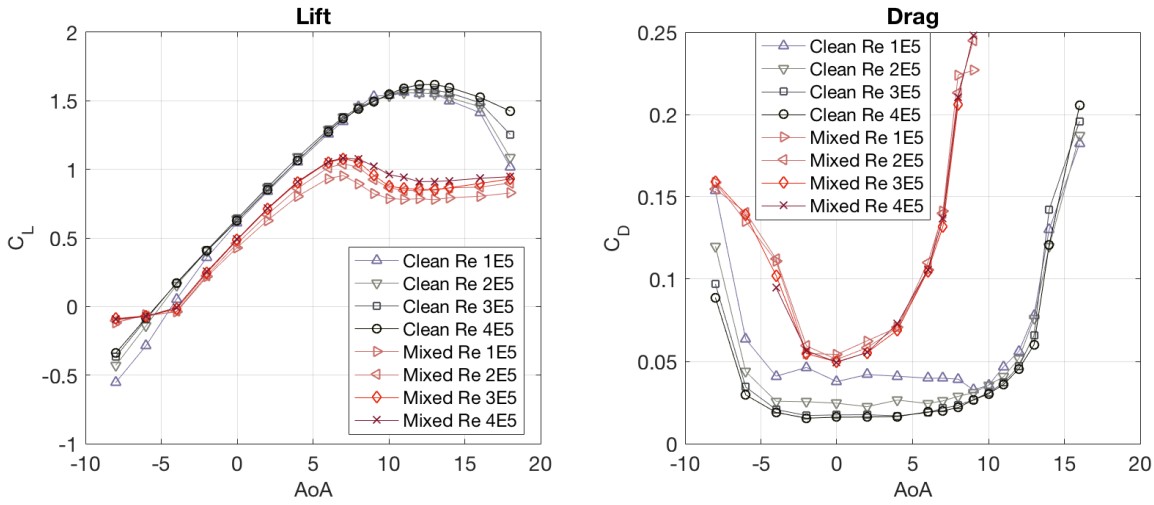

**Figure 8.** $C_L$- and $C_D$ curves for Reynolds numbers $1 \times 10^5, 2 \times 10^5, 3 \times 10^5$ and $4 \times 10^5$

### 4   Conclusions

A combined experimental and computational study on the NREL S826 airfoil was conducted to analyze the aerodynamic effects of leading-edge ice accretion. Three representative ice shapes were 3D printed from models made in LEWICE ice prediction tool. Experimental results were compared with computational analyses conducted in FENSAP for the same airfoil.

Following are our key conclusions:

- All three ice shapes resulted in reduced lift and increased drag compared with the airfoil without icing. Differences increased after onset of trailing edge separation.



- The rime- and glaze ice shapes investigated had a similar impact on the performance, both quantitatively and qualitatively. In the typical operating range, lift was reduced by 10% and drag increased by 80%.

- Mixed ice, with its horn-like shape, had a more severe impact on the aerodynamic behavior of the airfoil. Stall was reached at a 5° lower AoA than the clean airfoil. In the area before stall, lift was reduced by 30% and drag was increased by 340%.

- All ice types lead to performance losses of a magnitude that will reduce power output significantly.

- Simulation results show good agreement for the different test cases. The deviations are most pronounced after stall, leading to the assumption that they could be related to inaccuracies in the turbulence model and the absence of a transition model. Experimental measurements, especially for drag, are less reliable after stall, which is assumed to contribute to the deviations.

- The resemblance in tendencies to experimental results seen in this study show that there is great potential in applying CFD icing methods to investigate cold climate impacts on wind turbines.

- Further investigation on impact of ice extent, both in span and chord directions, would provide useful insight to the total effect of icing on a wind turbine installation.

*Competing interests.* The authors declare that they have no conflict of interest.





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
