# Peer review of "Aerodynamic Performance of the NREL S826 Airfoil in Icing Conditions"

_Wind Energy Science, 2017_

## Referee Comment (RC1) · Anonymous Referee #1 · 16 Nov 2017

The manuscript presents an experimental and a numerical study on the performance analysis of a S826 airfoil profile under icing conditions. Ice profiles predicted by LEWICE are 3-D printed and attached to the leading edge of a model for the experimental study. Rime, glaze and mixed type ice profiles are considered. In wind tunnel experiments the aerodynamic loads at various angle of attacks and Reynolds numbers are measured. A commertial flow solver FENSAP-ICE and an open source panel code, XFOIL, are used for the numerical solutions.

The major weakness of the study:

1- It is a performance analysis based on 2D, steady flows. Similar studies were already performed and published in the past such as:

W. J. Jasinski, S.c. Noe, M.S. Selig and M. B. Bragg., Wind Turbine Performance Under Icing Conditions, Journal of Solar Engineering, Vol.120, pp. 60-65, Feb 1998.

Bragg, M.B, Broeren, A.P., Andy H.E., Potapczuk, M.G., Guffond D. and Montreuil E., Airfoil Ice-Accretion Aerodynamics Simulation, NASA/TM—2008-214830, Jan 2008

The current state of the art in research on icing of airfoils is more on 3D unsteady flow simulations and accurate predictions of ice accreation, and power losses.

In addition;

2- The study employs commercial or well-known open source tools which are deveoped in 1980s, and do not need validation. It does not help the objective of the study.

3- It is stated that "for the sake of simplicity, LEWICE was used for the ice generation and FENSAP only as a flow field solver.." FENSAP-ICE is a newer and more advanced approach to icing. It is not clear how such a choice serves the main objective of the study: "to obtain more knowledge about the effects of different ice accretions.."

4- The ice shapes given in Fig 1 are all mixed-up. The horn-ice shape in red should be the glaze ice, the smooth one in green is the rime ice and the blue is the mixed type.

5- "airfoil coefficients" used throughout the manuscript is a misnomer. It should be properly addressed as "aerodynamic force coefficients"..

In conclusion, the study, which is performed with well known tools and is based on 2D steady flows, does not contribute much to the state of the art in aerodynamics of iced turbine blades...

---

## Referee Comment (RC2) · Anonymous Referee #2 · 28 Nov 2017

**Review**

**General comments**

The authors present an experimental/computational study on the influence of ice accretion on the NREL S826 airfoil. The analysis considers steady and bidimensional conditions. The issue is relevant but several major modifications and additional work are proposed before taking a decision to consider the manuscript for publication.

I have three main concerns:

- In my opinion (and if you consider that I am wrong, please comment on that) the analysis of the influence of ice accretion on the aerodynamic performance of an airfoil has relevant stochastic characteristics. I suppose that the ice shapes, for given conditions (air speed and temperature), can vary, following a stochastic pattern, and therefore the disturbed $c_l(\alpha)$ and $c_d(\alpha)$ functions are stochastic functions. The authors should consider the possibility to do some statistic analysis by generating different ice shapes and the corresponding $c_l(\alpha)$ and $c_d(\alpha)$ functions. In other words, how representative are the modified $c_l(\alpha)$ and $c_d(\alpha)$ functions that you have obtained since they correspond to a single test case?. I understand that testing several models in the wind tunnel is expensive but once you have confidence in your RANS model, to do numerical test is really cheap.

- Two relevant airfoil parameters when analysing the optimum performance of a wind turbine are the airfoil aerodynamic efficiency, $k(\alpha)$, and the optimum angle of attack, $\alpha_{op}$. There is not any comment on these parameters. The authors should analyse these parameters.

- Transforming the modification in $c_l(\alpha)$ and $c_d(\alpha)$ (and therefore $k(\alpha)$) into changes in the $C_P(\lambda, \theta_C, \mathrm{Re})$ map of the rotor is very relevant, and it is even more relevant when the authors continuously refer to the effect of the modification in $c_l(\alpha)$ and $c_d(\alpha)$ on the rotor performance and the authors refer to such type of quantification from other authors. They should include their own quantification.

**Abstract**

"...with a lift reduction of up to 30% in the linear lift area..." by "...with a lift reduction of up to 30% in the linear lift coefficient region...".

What about aerodynamic efficiency and optimum angle of attack?.

**1.1. Background**

I miss a reference to EU project WIND ENERGY PRODUCTION IN COLD CLIMATE (WECO).

"For wind turbine operation, there are several problems related to icing." by "There are several problems related to icing for wind turbine operation."

"Increased risk of structural fatigue,..." by "Increased levels of structural fatigue,..."

"...safety hazards of ice throw, electrical and mechanical failures..." by "safety hazards of ice throw and electrical and mechanical failures...".

"...Icing can also lead to overproduction and thus excessive structural loads, due to sudden increase in momentum, which the wind turbine is not dimensioned for (Jasinski et al., 1998)." Please explain a bit more (an additional sentence is enough). Is the mentioned overproduction due to increased air density?

"Further development in computational fluid dynamics, through experimental validation, will make information more available and less expensive to obtain when evaluating new challenges.". Too ambiguous sentence. What are the mentioned new challenges?.

**1.2. Objective**

"Their effects on aerodynamic properties were investigated." Please be more specific, describe from the very beginning which parameters will be analysed.

**2. Method**

"The blade tip is the part that is most exposed to icing due to large tip velocities leading to high accumulation rates." Since you are presenting a very general description of the ice accumulation phenomenon, a reference on this issue would be welcome here.

"...a hydraulically smooth surface was used for the experiments (Aksnes, 2015)". Have you calculated any kind of friction Reynolds number to state that your surface is hydraulically smooth?.

**2.1. Defining Icing Cases**

"...in the code are not excluding low-Reynolds numbers...". Please indicate here what is the range of Reynolds numbers in your test.

"...therefor..." by "...therefore...".

"...$t_{icing}$..." by "...$t_i$...". Check the whole document.

"Three ice shapes were selected, which are mainly distinguished by the temperature at which they form." This process probably has a high stochastic nature, so that, for given values of $V_\infty$ and $T$, the ice shapes are different from one experiment to another. This stochastic nature should be considered, if not, you are presenting a single case study that could no be representative (in the statistical sense).

Table 1. $V_\infty = 25$m/s, 40m/s values seem to be small compared with typical blade tip speed values. Why these small values are chosen?

"The surface roughness ks for each icing case..." is $k_s$ the equivalent sand-grain roughness? please clarify.

**2.2. Experimental Setup**

"$A_{wing}/A_{tunnel}$" by "$A_W/A_T$".

Figure 2. Enhance quality of text and lines on the figure (a).

**2.3. Measurement Methods.**

"To determine the lift and drag of the wing, force- and pressure measurements were applied." this isolated sentence here does not fit well.

"For all test cases, including with icing, the area was calculated using the clean wing chord length." Is this precise enough? you are not measuring a 2% or 5% of the chord where high suction values can occur. Also the real chord is 2% to 5% larger, what means differences in $c_d$ of this order of magnitude for the different ice/no ice cases (this is the order of the differences that you are presenting in your figures for $c_d$ in the linear region for $c_l$. Please comment on this.

"Normalized pressure, Cp, was..." by "Pressure coefficient, $C_p$, was..."

"...obtained surface pressure distributions exclude the measurements from these taps." Comment of the related error (see my comment at the beginning of this section).

Expression (3). Please include the integration limits.

"...velocity and y is the width..." by "...velocity and $y$ is the width..."

Expressions (4) and (5) not in line. $p_0$ is clear in the mentioned expressions but which speed is $u$?.

"This means up to around 12º for the clean airfoil, and less for the icing cases depending on the shape." The term "less" is too vague, please be more specific.

"Uncertainty estimations show an offset of approximately $C_D = 0.01$ in the calculated drag, over the applied range of AoA, due to static pressure effects." Please quantify the associated uncertainty in the maximum aerodynamic efficiency.

**2.4. Simulation Setup.**

"...a state-of-the-art Navier-Stokes CFD solver..." Are you using RANS, LES? please clarify from the very beginning.

**3. Results and Discussion**

"However, the objective of this study is to discover general trends of icing impact on aerodynamic performance, so the Re mismatch is considered to be not significant." I would not state that since your are testing in a Reynolds number region where the aerodynamic coefficients are quite sensitive to the variation of this non-dimensional number. So, I would say that the influence of Reynolds number mismatch deserves further research efforts that are out of the scope of this work.

"In the linear area..." by "In the linear region..."

"The current study shows that mixed ice causes more severe performance losses than rime- and glaze ice, leading to the assumption that less streamlined ice shapes can reduce power output even more." Why not concluding on variations of $c_l$, $c_d$ and $k$ instead of directly concluding on the effects on the wind turbine power output, without any calculation?. I would compare your results on variations of $c_l$, $c_d$ and $k$ with the results of other authors, and then, and only then, I would translate my conclusions to the influence on the rotor performance parameters (power coefficient, equivalent loads...) after performing my own aeroelasic analysis of the rotor equipping the ice covered airfoils.

"Mixed ice is,..." by "Mixed ice case is,..."

"In this area, the..." by "In this region, the..."

"...ice curves start to incline..." please rephrase this sentence.

"The lift decrease and drag increase that can be observed for all ice shapes indicate that icing generally leads to reduced performance and hence power output losses." It is true, but is too vague, quantify the variation in the aerodynamic efficiency and then in power coefficient.

Figure 3. Include similar figures for the aerodynamic efficiency. You are showing differences for $c_d$ in the linear $c_l$ region less than 0.01 which is the uncertainty of your $c_d$ measurements due to $p_0$ hypothesis. You should consider the bias associated to consider the clean airfoil chord value to calculate coefficients for all the cases.

**3.2 Prediction of Airfoil Coefficients by FENSAP-ICE**

"Computational lift values show good..." by "Lift coefficient values provided by the computational model..."

..."experimental values in the linear area..." by "experimental values in the linear region..."

"...while it deviates more around stall occurrence..." by "...while larger differences are found in the stall region..."

"The curves, however, display a large resemblance." by "Trends are well predicted.".

"The computational drag values follow..." by "The drag values predicted by the computational model follow..."

"...for all the icing cases before stall occurs..." by "...for all the icing cases out of the stall region..."

"Deviations in this area is expected,..." by "Deviations in this region are expected,..."

"The deviations seen..." by "The deviations observed..."

"The Spalart-Allmaras turbulence..." by "Firstly, the Spalart-Allmaras turbulence..."

"...assumes fully turbulent flow." by "...assumes fully turbulent flow, and therefore laminar-turbulent transition is not predicted".

"For the complex shapes studied, at AoAs..." by "Secondly, for the complex shapes studied, at AoAs..."

"...the aerodynamic characteristics are affected in ways that are not necessarily captured by a simplified turbulence model." Too vague, please specify a bit more. Include a relevant reference on this issue.

**3.3 Surface Pressure Distributions**

"...lift vulnerable to disturbances in this area." by "...lift vulnerable to disturbances in this region".

"...over the ice shape." by "on the ice shape on the upper side of the airfoil."

"...by the relatively constant $C_P$ in this space..." by "...by the relatively constant $C_p$ in this region..."

About the comparison between computational and experimental results. Since you quantify differences between wind tunnel and computational results in your analysis, I recommend to present a figure showing such differences instead of forcing the reader to calculate these differences on line.

"One reason for this is likely the earlier onset of trailing edge separation on the suction side when icing is present, making it difficult to predict pressure correctly by the turbulence model." This is a bit confusing considering your previous arguments. You stated that the weak point of your RANS model is that it considers fully turbulent conditions and, therefore, transition is not predicted. Now, when ice formations exist guarantying fully turbulent conditions downstream the leading edge, you state that this is a weak point of your model. Please clarify a bit

more what are the weak points of your computational model. For sure there are excellent works of other authors analysing the drawbacks of RANS models when predicting flow behaviour around airfoils. I recommend to refer to them.

"All three icing shapes are likely to trigger laminar..." by "All three icing shapes are likely to trigger laminar-turbulent..."

Why do you add results from XFOIL?, what is the contribution?

Check formats: "$x/c$" instead of x/c...Please review the whole paper.

What does "SP" mean?

"...it is not able to capture laminar transitions..." by "...laminar-turbulent transitions are not captured..."

"In addition, low flow velocities experience higher relative disturbances, adding uncertainty to the measurements." Explain a bit more. Add a reference.

A really linear $c_l(\alpha)$ region is not detected in your figure 8 (there is a clearly detectable curvature in the $c_l(\alpha)$ curve for $\alpha \in [-5, 5]$deg. Why? Compare with other authors [1]. Is the S806 airfoil an airfoil not exhibiting a linear $c_l(\alpha)$ region?.

**3.4 Reynolds Number Dependency**

"Ice accretion show..." by "Ice accretion shows..."

"...which will lead to rotational losses.." What are "rotational losses"?

The influence of Re is really small for the range of Re values explored. Please compare your results with results of other authors, other airfoils.

**4. Conclusions**

"The rime- and glaze ice shapes investigated had a similar impact on the performance, both quantitatively and qualitatively. In the typical operating range, lift was reduced by 10% and drag increased by 80%." In which $\alpha$ range?

"In the area before stall, lift..." by "In the regions before stall, lift..."

"In the area before stall, lift was reduced by 30% and drag was increased by 340%." specify $\alpha$ range.

"All ice types lead to performance losses of a magnitude that will reduce power output significantly." Please use quantitative conclusions. Calculate the effect on aerodynamic efficiency and power coefficient if you pretend to present conclusions on the influence in the rotor performance.

"The deviations are most pronounced after stall, leading to the assumption that they could be related to inaccuracies in the turbulence model and the absence of a transition model." Please support this conclusion on an evidence or on the work of other authors.

"...great potential in applying CFD icing methods..." CFD icing method is confusing, please rephrase the sentence.

"Further investigation on impact of ice extent, both in span and chord directions, would provide useful insight to the total effect of icing on a wind turbine installation." Vague conclusion. Please be more specific, what about the stochastic nature of the ice formation process?.

**References**

[1]. H. Sarlak , R. Mikkelsen, S. Sarmast, JN. Sφrensen, Aerodynamic behaviour of NREL S826 airfoil at Re=100,000, Journal of Physics: Conference Series 524 (2014) 012027

---

## Author Comment (AC1) · 7 Jan 2018

**Authors' response on referee comments on « Aerodynamic Performance of the NREL S826 Airfoil in Icing Conditions» by Julie Krøgenes et al.**

**J. Krøgenes et al.**
juliekrogenes@gmail.com

We thank the referees for their critical and appropriate comments. We were asked to answer all referee comments at this stage of the review process, while a revised manuscript should not be prepared at this stage yet. In the following, we will therefore engage with all the comments and propose improvements for the final manuscript.

**Reply to Anonymous Referee #1 (RC1):**

**Comment RC1-1:**
Similar studies were already performed and published in the past such as: C1 WESD Interactive comment Printer-friendly version Discussion paper W. J. Jasinski, S.c. Noe, M.S. Selig and M. B. Bragg., Wind Turbine Performance Under Icing Conditions, Journal of Solar Engineering, Vol.120, pp. 60-65, Feb 1998. Bragg, M.B, Broeren, A.P., Andy H.E., Potapczuk, M.G., Guffond D. and Montreuil E., Airfoil Ice-Accretion Aerodynamics Simulation, NASA/TM T2008-214830, Jan 2008 ˘ The current state of the art in research on icing of airfoils is more on 3D unsteady flow simulations and accurate predictions of ice accretion, and power losses.

**The authors' reply to RC1-1:**
We acknowledge that similar work has been performed in the past such as listed by the reviewer. However, there is a significant difference in the Reynolds-number regime. In the current literature, all wind tunnel experiments are performed at Re>1e6, whereas this study focuses on the low-Re regime Re<5e5. At such low Reynolds number transitional effects of the boundary layer add additional complexity to the problem. Therefore, icing experiment at these low Re numbers is important to extend the validity of the icing tools as well as for special applications such as small wind turbines or UAVs.

**Comment RC1-2:**
The study employs commercial or well-known open source tools which are developed in 1980s, and do not need validation. It does not help the objective of the study.

**The authors' reply to RC1-2:**
As stated on page 3, line 3 of the manuscript (Ref Wright, 1999), LEWICE is not validated for Reynolds numbers below 2.26e6.

**Comment RC1-3:**
It is stated that "for the sake of simplicity, LEWICE was used for the ice generation and FENSAP only as a flow field solver.." FENSAP-ICE is a newer and more advanced approach to icing. It is not clear how such a choice serves the main objective of the study: "to obtain more knowledge about the effects of different ice accretions.."

**The authors' reply to RC1-3:**
Generating complex ice shapes with FENSAP-ICE is a very time-consuming and labor-intensive effort. The idea of this paper was to find representative ice shapes for different icing cases. In order to obtain these, several hundred of parameter combinations (LWC, MVD, Temp, icing duration, velocity) have been studied. For investigating such a large number of cases, LEWICE is the tool of choice as FENSAP calculations would take unreasonably long

time. Furthermore, the main objective has been refocused on the topic of obtaining more knowledge about the effects of different ice accretions *at low Reynolds numbers*.

**Comment RC1-4:**
The ice shapes given in Fig 1 are all mixed-up. The horn-ice shape in red should be the glaze ice, the smooth one in green is the rime ice and the blue is the mixed type.

**The authors' reply to RC1-4:**
The ice shapes were generated using LEWICE, and these are the ice shapes predicted for the temperature ranges characterizing rime-, glaze- and mixed ice.

**Comment RC1-5:**
"airfoil coefficients" used throughout the manuscript is a misnomer. It should be properly addressed as "aerodynamic force coefficients"..

**The authors' reply to RC1-5:**
The suggested formulation is more precise and will be changed accordingly.

**In general:**
In conclusions, we thank the reviewer for his/her comments. The reviewer highlighted some wording issues and wrong figure labels. More generally his/her comments helped us realize that the objective of the paper needs to be more specific. Following the reviewer's argumentation, we will specify in the introduction how these experiments have been performed for low Reynold numbers which have not been previously covered by the literature. To emphasize the objective of the paper the authors would like to propose a change in the title to "Low-Reynolds Aerodynamic Performance of the NREL S826 Airfoil in Icing Conditions".

---

## Author Comment (AC2) · 7 Jan 2018

**Authors' response on referee comments on « Aerodynamic Performance of the NREL S826 Airfoil in Icing Conditions» by Julie Krøgenes et al.**

**J. Krøgenes et al.**

juliekrogenes@gmail.com

We thank the referees for their critical and appropriate comments. We were asked to answer all referee comments at this stage of the review process, while a revised manuscript should not be prepared at this stage yet. In the following, we will therefore engage with all the comments and propose improvements for the final manuscript.

**Reply to Anonymous Referee #2 (RC2):**

**Comment RC2-1:**
In my opinion (and if you consider that I am wrong, please comment on that) the analysis of the influence of ice accretion on the aerodynamic performance of an airfoil has relevant stochastic characteristics. I suppose that the ice shapes, for given conditions (air speed and temperature), can vary, following a stochastic pattern, and therefore the disturbed $cl(\alpha)$ and $cd(\alpha)$ functions are stochastic functions. The authors should consider the possibility to do some statistic analysis by generating different ice shapes and the corresponding $cl(\alpha)$ and $cd(\alpha)$ functions. In other words, how representative are the modified $cl(\alpha)$ and $cd(\alpha)$ functions that you have obtained since they correspond to a single test case? I understand that testing several models in the wind tunnel is expensive but once you have confidence in your RANS model, to do numerical test is really cheap.

**The authors' reply to RC2-1:**
We agree that stochastics could be relevant and may be worth further investigation. However, the objective of this work is to perform an experimental validation at low Reynolds numbers. Therefore, three defined typical icing cases have been selected and tested experimentally and numerically. Our 3 different profiles are the results of what is obtained using icing prediction tools normally being used in engineering. As there is a stochastic variation of these cases (combinations of the investigated shapes and more extreme shapes), our results can be used in a future numerical setup to investigate the sensitivity to ice shape variations. Based on the literature, the authors are not aware of any stochastic approach to this issue.

**Comment RC2-2:**
Two relevant airfoil parameters when analysing the optimum performance of a wind turbine are the airfoil aerodynamic efficiency, $k(\alpha)$, and the optimum angle of attack, $\alpha_{op}$. There is not any comment on these parameters. The authors should analyse these parameters.

**The authors' reply to RC2-2:**
We acknowledge that it might be relevant to investigate the impact of icing on the aerodynamic efficiency and possible changes in $\alpha\_opt$. However, we are only focussing on a 2D airfoil data for validation purposes in this study. While the aerodynamic efficiency and $\alpha\_opt$ are important parameters for the analysis of an (iced) wind turbine rotor, we do not see the added benefit for a 2D exp-sim comparison of the airfoil's force coefficients.

**Comment RC2-3:**
Transforming the modification in cl(α) and cd(α) (and therefore k(α)) into changes in the CP (λ, θC , Re) map of the rotor is very relevant, and it is even more relevant when the authors continuously refer to the effect of the modification in cl(α) and cd(α) on the rotor performance and the authors refer to such type of quantification from other authors. They should include their own quantification.

**The authors' reply to RC2-3:**
We acknowledge that it might be an added benefit to investigate the impact of icing on a wind turbine blade at low Reynolds-numbers. However, at this stage we are only focussing on a 2D airfoil data for validation purposes. Our 2D results cannot be used to generate representative data for an iced turbine blade (e.g. by using BEM methods). The main reason for this is that icing does not occur homogeneously alongside a wind turbine blade. The ice accretion process is heavily governed by local wind velocities and chord sizes, which means that the ice geometries will vary significantly alongside the blade (short ice horns near the hub and large ice horns near the tip). Hence our results for a single size of the horn (but different icing types) is not suited for this kind of investigation.

**Comment RC2-4:**
"...with a lift reduction of up to 30% in the linear lift area..." by "...with a lift reduction of up to 30% in the linear lift coefficient region...". What about aerodynamic efficiency and optimum angle of attack?

**The authors' reply to RC2-4:**
The suggested formulation is more precise and will be changed accordingly. Regarding the aerodynamic efficiency and optimum angle of attack, we do not see the added benefit of including these in a 2D exp-sim comparison of the airfoil's force coefficients, as stated in **RC2-2**.

**Comment RC2-5:**

I miss a reference to EU project WIND ENERGY PRODUCTION IN COLD CLIMATE (WECO).

**The authors' reply to RC2-5:**
The report on the suggested project WIND ENERGY PRODUCTION IN COLD CLIMATE (WECO) states that experimental results and numerical simulations show that attached ice on wind turbine rotor blades or other components lead to decreased production, and will be a valuable reference in the paper's introduction.

Ref. Tammelin, B. et al.: WIND ENERGY PRODUCTION IN COLD CLIMATE (WECO), Finnish Meteorological Institute,  JOR3-CT95-0014, 1998

**Comment RC2-6:**
"For wind turbine operation, there are several problems related to icing." by "There are several problems related to icing for wind turbine operation." "Increased risk of structural fatigue..." by "Increased levels of structural fatigue,..." "...safety hazards of ice throw, electrical and mechanical failures..." by "safety hazards of ice throw and electrical and mechanical failures...".

**The authors' reply to RC2-6:**
The suggested formulations are more precise and will be changed accordingly.

**Comment RC2-7:**
"...Icing can also lead to overproduction and thus excessive structural loads, due to sudden increase in momentum, which the wind turbine is not dimensioned for (Jasinski et al., 1998)." Please explain a bit more (an additional sentence is enough). Is the mentioned overproduction due to increased air density?

**The authors' reply to RC2-7:**
We thank the reviewer for this comment, which made us realize that this statement adds no value to the scope of the paper. It is true that higher air density at low temperatures can lead to a higher production. In addition, some cases of ice accretion can act as a leading edge flap increasing lift and delaying stall. However, this is very rare and the sentence mentioned will therefore be removed from the paper.

**Comment RC2-8:**
"Further development in computational fluid dynamics, through experimental validation, will make information more available and less expensive to obtain when evaluating new challenges.". Too ambiguous sentence. What are the mentioned new challenges?

**The authors' reply to RC2-8:**
By new challenges the authors refer to future evaluations of wind turbine icing in general. The authors agree that the sentence is too ambiguous, it will therefore be rephrased as follows: "Further development in computational fluid dynamics, through experimental validation, will make information more available and less expensive to obtain for further icing evaluation."

**Comment RC2-9:**
"Their effects on aerodynamic properties were investigated." Please be more specific, describe from the very beginning which parameters will be analysed.

**The authors' reply to RC2-9:**
The authors will include a more specific description of which parameters that were analysed in the revised paper.

**Comment RC2-10:**
"The blade tip is the part that is most exposed to icing due to large tip velocities leading to high accumulation rates." Since you are presenting a very general description of the ice accumulation phenomenon, a reference on this issue would be welcome here.

**The authors' reply to RC2-10:**
The following reference on this issue will be included:
O. Parent, A. Ilinca, Anti-icing and de-icing techniques for wind turbines: Critical review, Cold Regions Science and Technology, vol 65, issue 1, pages 85-96, 2011

**Comment RC2-11:**
"...a hydraulically smooth surface was used for the experiments (Aksnes, 2015)". Have you calculated any kind of friction Reynolds number to state that your surface is hydraulically smooth?.

**The authors' reply to RC2-11:**
The wing model's surface roughness profile was measured with a digitizing arm and assessed to be smaller than the viscous sublayer thickness. According to [Pope, 2000] the roughness will in this case not affect the skin friction and the flow. The mean surface roughness was measured to be 0.78 micrometers, while the viscous sublayer thickness was calculated to be 16 micrometers (for Re=100.000) [Pope, 2000]. Therefore, the surface layer could be assessed to be hydraulically smooth.

Reference:
Pope, S.B.: Turbulent Flows, Cambridge University Press, 2000.

**Comment RC2-12:**
"...in the code are not excluding low-Reynolds numbers...". Please indicate here what is the range of Reynolds numbers in your test.

**The authors' reply to RC2-12:**
The range of Reynolds numbers included in the tests are Re = 1e5, 2e5, 3e5 and 4e5 and will be included in the revised version of the paper.

**Comment RC2-13:**
"...therefor..." by "...therefore...".
 "...ticing..." by "...ti ...". Check the whole document.

**The authors' reply to RC2-13:**
This will be corrected and replaced throughout the document.

**Comment RC2-14:**
"Three ice shapes were selected, which are mainly distinguished by the temperature at which they form." This process probably has a high stochastic nature, so that, for given values of $V\infty$ and T, the ice shapes are different from one experiment to another. This stochastic nature should be considered, if not, you are presenting a single case study that could no be representative (in the statistical sense).

**The authors' reply to RC2-14:**
The section will be rephrased: "The main objective of this study is to perform icing experiments and a validation of the prediction tool FENSAP-ICE at a low Reynolds regime on three selected icing cases. It should be noted that in reality the formation of ice shapes is a stochastic process. However, when using simulation methods such as LEWICE or FENSAP-ICE, it is assumed that the resulting ice shapes are representative for the stochastic variability."

**Comment RC2-15:**
Table 1. $V\infty$ = 25m/s, 40m/s values seem to be small compared with typical blade tip speed values. Why these small values are chosen?

**The authors' reply to RC2-15:**
The velocity values were chosen based on the consideration of the (low) Reynolds number and in order to obtain representative icing geometries. The velocity of 40m/s was specifically chosen in order to generate the horn (or lobster-tail) ice geometries.

**Comment RC2-16:**
"The surface roughness ks for each icing case..." is ks the equivalent sand-grain roughness? please clarify.

**The authors' reply to RC2-16:**
Yes, this is correct and will be clarified. The sentence will be rephrased as follows: "The equivalent sand grain roughness ks for each icing case…"

**Comment RC2-17:**
"Awing/Atunnel" by "AW /AT ". Figure 2. Enhance quality of text and lines on the figure (a).

**The authors' reply to RC2-17:**
The suggested formulation is more precise and will be changed accordingly. The quality of figure (a) will be enhanced.

**Comment RC2-18:**
"To determine the lift and drag of the wing, force- and pressure measurements were applied." this isolated sentence here does not fit well.

**The authors' reply to RC2-18:**
The authors agree, and the sentence will be removed as it does not contribute with additional information.

**Comment RC2-19:**
"For all test cases, including with icing, the area was calculated using the clean wing chord length." Is this precise enough? you are not measuring a 2% or 5% of the chord where high suction values can occur. Also the real chord is 2% to 5% larger, what means differences in cd of this order of magnitude for the different ice/no ice cases (this is the order of the differences that you are presenting in your figures for cd in the linear region for cl . Please comment on this.

**The authors' reply to RC2-19:**
The authors believe that referring always to the clean wing chord length is the most coherent way of presenting the results. The reason is that it is considered more useful to show the changes the ice accretion gives related to the clean case. Also, a definition of a new chord length with the presence of ice on the surface has not been found in the literature, showing that it is common to use the chord for clean condition also in the presence of ice (see e.g. Broeren, 2011). For the same reason the AoA is defined by referring to the chord length in clean condition.

Nevertheless, the average ice roughness height divided by the chord length will be reported in table 1 and a comment on the change in geometry and consequence on the results will be included in the revised version of the paper.

Ref. Broeren, A. P., et al., Aerodynamic simulation of ice accretion on airfoils, 2011

**Comment RC2-20:**
"Normalized pressure, Cp, was..." by "Pressure coefficient, Cp, was..."

**The authors' reply to RC2-20:**
The suggested formulation is more precise and will be changed accordingly.

**Comment RC2-21:**
"...obtained surface pressure distributions exclude the measurements from these taps."
Comment of the related error (see my comment at the beginning of this section).

**The authors' reply to RC2-21:**
The related error will be discussed in the revised paper, as stated in reply RC2-19 .

**Comment RC2-22:**
Expression (3). Please include the integration limits. "...velocity and y is the width..." by
"...velocity and y is the width..."

**The authors' reply to RC2-22:**
The authors agree and the suggested changes will be implemented.

**Comment RC2-23:**
Expressions (4) and (5) not in line. p0 is clear in the mentioned expressions but which speed
is u?

**The authors' reply to RC2-23:**
Earlier in the mentioned section it has been specified which speed u is. In order to make it
clearer, the authors will include an explanation below the expressions. In the author's version
of the manuscript, the expressions seem to be in line. This will be double checked.

**Comment RC2-24:**
"This means up to around 12º for the clean airfoil, and less for the icing cases depending on
the shape." The term "less" is too vague, please be more specific

**The authors' reply to RC2-24:**
The authors agree and will rephrase the sentence as follows:
"This means up to around 12º for the clean airfoil, and between 7º and 12º for the icing cases
depending on the shape."

**Comment RC2-25:**
"Uncertainty estimations show an offset of approximately CD = 0.01 in the calculated drag,
over the applied range of AoA, due to static pressure effects." Please quantify the associated
uncertainty in the maximum aerodynamic efficiency.

**The authors' reply to RC2-25:**
The authors agree that the uncertainty on $C_D$ due to the static pressure, for wake rake has to
be quantified. This can be done referring to previous experiments where the results with and
without static pressure have been investigated. The paragraph could be rewritten as follows:

Based on previous studies (Bracchi, 2014) it is expected a higher $C_D$ when including the
wake static pressure in the measurements, with a maximum increment of 12%. Hence the
maximum uncertainty on $C_D$ for the range of AoAs with attached flow is estimated to be
maximum 12%.

*Ref. Bracchi, T: Downwind Rotor: Studies on yaw Stability and Design of a Suitable Thin Airfoil, 2014.*

**Comment RC2-26:**
"...a state-of-the-art Navier-Stokes CFD solver..." Are you using RANS, LES? please clarify from the very beginning.

**The authors' reply to RC2-26:**
The Navier-Stokes RANS CFD solver has been used, and we will change this notation in the revised version.

 **Comment RC2-27:**
"However, the objective of this study is to discover general trends of icing impact on aerodynamic performance, so the Re mismatch is considered to be not significant." I would not state that since your are testing in a Reynolds number region where the aerodynamic coefficients are quite sensitive to the variation of this non-dimensional number. So, I would say that the influence of Reynolds number mismatch deserves further research efforts that are out of the scope of this work.

**The authors' reply to RC2-27:**
The authors agree and will implement the suggested changes.

**Comment RC2-28:**
"In the linear area..." by "In the linear region..."

**The authors' reply to RC2-28:**
The suggested formulation is more precise and will be changed accordingly.

**Comment RC2-29:**
"The current study shows that mixed ice causes more severe performance losses than rime- and glaze ice, leading to the assumption that less streamlined ice shapes can reduce power output even more." Why not concluding on variations of cl , cd and k instead of directly concluding on the effects on the wind turbine power output, without any calculation?. I would compare your results on variations of cl , cd and k with the results of other authors, and then, and only then, I would translate my conclusions to the influence on the rotor performance parameters (power coefficient, equivalent loads...) after performing my own aeroelastic analysis of the rotor equipping the ice covered airfoils.

**The authors' reply to RC2-29:**
The authors agree that concluding on variations on $C_L$, $C_D$ and lift-to-drag ratio is a better way than discussing change in power. The paragraph will be rewritten accordingly.

**Comment RC2-30:**
"Mixed ice is,..." by "Mixed ice case is,..." "In this area, the..." by "In this region, the..."

**The authors' reply to RC2-30:**
The suggested formulations are more precise and will be changed accordingly.

**Comment RC2-31:**
"...ice curves start to incline..." please rephrase this sentence.

**The authors' reply to RC2-31:**
The authors agree and the sentence will be changed to: "Rime- and glaze ice curves have a shorter constant drag range, while the mixed ice curve shows no constant behaviour."

**Comment RC2-32:**
"The lift decrease and drag increase that can be observed for all ice shapes indicate that icing generally leads to reduced performance and hence power output losses." It is true, but is too vague, quantify the variation in the aerodynamic efficiency and then in power coefficient.

**The authors' reply to RC2-32:**
As stated in **RC2-3**, aerodynamic efficiency and power coefficient are parameters for a wind turbine blade that we do not investigate in this paper.

**Comment RC2-33:**
Figure 3. Include similar figures for the aerodynamic efficiency. You are showing differences for cd in the linear cl region less than 0.01 which is the uncertainty of your cd measurements due to p0 hypothesis. You should consider the bias associated to consider the clean airfoil chord value to calculate coefficients for all the cases.

**The authors' reply to RC2-33:**
As stated in **RC2-3**, aerodynamic efficiency and power coefficient are parameters for a wind turbine blade that we do not investigate in this paper.

**Comment RC2-34:**
"Computational lift values show good..." by "Lift coefficient values provided by the computational model..." ..."experimental values in the linear area..." by "experimental values in the linear region..." "...while it deviates more around stall occurrence..." by "...while larger differences are found in the stall region..." "The curves, however, display a large resemblance." by "Trends are well predicted.". "The computational drag values follow..." by "The drag values predicted by the computational model follow..." "...for all the icing cases before stall occurs..." by "...for all the icing cases out of the stall region..." "Deviations in this area is expected…" by "Deviations in this region are expected…" "The deviations seen..." by "The deviations observed..." "The Spalart-Allmaras turbulence..." by "Firstly, the Spalart-Allmaras turbulence..." "...assumes fully turbulent flow." by "...assumes fully turbulent flow, and therefore laminar-turbulent transition is not predicted". "For the complex shapes studied, at AoAs..." by "Secondly, for the complex shapes studied, at AoAs..."

**The authors' reply to RC2-34:**
The suggested formulations are more precise and will be changed accordingly.

**Comment RC2-35:**
"...the aerodynamic characteristics are affected in ways that are not necessarily captured by a simplified turbulence model." Too vague, please specify a bit more. Include a relevant reference on this issue.

**The authors' reply to RC2-35:**
The authors agree that the mentioned formulation is too vague. The sentence will be changed to: "...a simplified turbulence model may not capture all aerodynamic effects as

shown by Bartl et al. (2018). Effects like laminar separation bubbles in transitional Regimes regimes have been shown not to be captured by a RANS k-epsilon turbulence model."

*Reference:*
*J. Bartl, K. Sagmo, T. Bracchi, L. Sætran (2018)*
*Performance of the NREL S826 airfoil at low to moderate Reynolds numbers: A reference experiment for CFD models.*
*Manuscript submitted to European Journal of Mechanics - B/Fluids*

**Comment RC2-36:**
"...lift vulnerable to disturbances in this area." by "...lift vulnerable to disturbances in this region". "...over the ice shape." by "on the ice shape on the upper side of the airfoil." "...by the relatively constant CP in this space..." by "...by the relatively constant Cp in this region..."

**The authors' reply to RC2-36:**
The suggested formulations are more precise and will be changed accordingly.

**Comment RC2-37:**
About the comparison between computational and experimental results. Since you quantify differences between wind tunnel and computational results in your analysis, I recommend to present a figure showing such differences instead of forcing the reader to calculate these differences on line.

**The authors' reply to RC2-37:**
The quantifications of differences between computation and experimental results are all referred to in presented figures. Differences in lift and drag refer to Figure 4, while differences in pressure coefficient Cp are presented in Figure 6. The quantifications of differences in percent are rough estimates of deviations in suctions side pressure level predictions. The description will be changed to a more precise formulation.

**Comment RC2-38:**
"One reason for this is likely the earlier onset of trailing edge separation on the suction side when icing is present, making it difficult to predict pressure correctly by the turbulence model." This is a bit confusing considering your previous arguments. You stated that the weak point of your RANS model is that it considers fully turbulent conditions and, therefore, transition is not predicted. Now, when ice formations exist guarantying fully turbulent conditions downstream the leading edge, you state that this is a weak point of your model. Please clarify a bit more what are the weak points of your computational model. For sure there are excellent works of other authors analysing the drawbacks of RANS models when predicting flow behaviour around airfoils. I recommend to refer to them.

**The authors' reply to RC2-38:**
The authors agree that the statement is misleading and contradicting previous statements. A clarification of the weak points will be added as well as additional references. The sentence will be rephrased: "The simulations does not seem to predict the onset of suction side separation correctly when compared with experimental results. Differences in the results could also likely be related to three-dimensional flow effects in the experiment at the onset of stall (Cakmakcioglu et al. (2014).

Reference:
S. C. Cakmakcioglu, I. O. Sert, O. Tugluk, N. Sezer-Uzol, 2-d and 3-d cfd
620 investigation of nrel s826 airfoil at low reynolds numbers, J. Phys.: Conf.

Series 524 (2014) 012028. doi:10.1088/1742-6596/524/1/012028.

**Comment RC2-39:**
Why do you add results from XFOIL?, what is the contribution?

**The authors' reply to RC2-39:**
XFOIL is a very reliable numeric tool for predicting airfoil performance at low Reynolds-numbers. This is due to the implementation of transition models which is lacking in our CFD runs. XFOIL has been included in order to highlight that the absence of laminar flow does not have a major impact.

**Comment RC2-40:**
Check formats: "x/c" instead of x/c...Please review the whole paper.

**The authors' reply to RC2-40:**
This correction will be implemented throughout the paper.

**Comment RC2-41:**
What does "SP" mean?

**The authors' reply to RC2-41:**
SP is an abbreviation for Surface Pressure integration. The authors will specify this throughout the paper in order to make it clearer.

**Comment RC2-42:**
"...it is not able to capture laminar transitions..." by "...laminar-turbulent transitions are not captured..."

**The authors' reply to RC2-42:**
The suggested formulation is more precise and will be changed accordingly.

**Comment RC2-43:**
"In addition, low flow velocities experience higher relative disturbances, adding uncertainty to the measurements." Explain a bit more. Add a reference.

**The authors' reply to RC2-43:**
The authors agree and the sentence will be changed to: "In addition, low flow velocities experience higher relative disturbances. For example, imperfections in the wing- and wind tunnel geometry will have a greater impact. In addition, low velocities are more vulnerable to small variations in the wind speed. This adds uncertainty to the measurements at low Re numbers."

**Comment RC2-44:**
A really linear cl($\alpha$) region is not detected in your figure 8 (there is a clearly detectable curvature in the cl($\alpha$) curve for $\alpha \in [-5, 5]$deg. Why? Compare with other authors [1]. Is the S806 airfoil an airfoil not exhibiting a linear cl($\alpha$) region?.

**The authors' reply to RC2-44:**
H. Sarlak et al. (2014) has tested a model of the NREL S826 airfoil at Re = 100 000. A comparison of the clean wing results with measurements by Sarlak et al. (2014) has been performed in another submitted paper by the authors (Bartl et al. (2018)), which shows good agreement in the linear lift region. A slight curvature is present in all performed experiments by Sarlak et al. (2014) and also Ostovan et al. (2013), indicating a not entirely linear lift of the

S826 in this region. The authors have discussed possible reasons for more unstable behavior at low Reynolds numbers in section 3.4 of the manuscript.

References:
J. Bartl, K. Sagmo, T. Bracchi, L. Sætran. Performance of the NREL S826 airfoil at low to moderate Reynolds numbers: A reference experiment for CFD models
Submitted to European Journal of Mechanics - B/Fluids (2018)

H. Sarlak, R. Mikkelsen, S. Sarmast, J. N. Sørensen, Aerodynamic behaviour of nrel s826 airfoil at re=100 000, J. Phys.: Conf. Series 524 (2014) 012027.

Y. Ostovan, H. Amiri, O. Uzol, Aerodynamic characterization of nrel s826 airfoil at low reynolds numbers, in: Conference on Wind Energy Science and Technology-RUZGEM, 2013.

**Comment RC2-45:**
"Ice accretion show..." by "Ice accretion shows..."

**The authors' reply to RC2-45:**
The spelling error will be corrected.

**Comment RC2-46:**
"...which will lead to rotational losses.." What are "rotational losses"?

**The authors' reply to RC2-46:**
By rotational losses the authors refer to induced drag due to streamwise vorticity. This will be specified in the revised version of the paper.

**Comment RC2-47:**
The influence of Re is really small for the range of Re values explored. Please compare your results with results of other authors, other airfoils.

**The authors' reply to RC2-47:**
Reports by Sarmast et al. (2013) and Ostovan et al. (2013) indicate a Reynolds independent behaviour of the S826 airfoil for Re>= 100.000, which is in good agreement with the presented results. A comment on a stronger Reynolds number dependency of other low Reynolds airfoils will be included in a revised version of the manuscript, e.g. Selig et al. (1995)

References:
S. Sarmast, R. Mikkelsen, The experimental results of the nrel s826 airfoil
at low reynolds numbers, Tech. rep., KTH (2013). URL: urn:nbn:se:kth:diva-120583

Y. Ostovan, H. Amiri, O. Uzol, Aerodynamic characterization of nrel s826 airfoil at low reynolds numbers, in: Conference on Wind Energy Science and Technology-RUZGEM, 2013.

Selig, M.S., Guglielmo, J.J., Broeren, A.P., and Giguere, P.: *Summary of Low-Speed Airfoil Data; Vol.1* SoarTech Publications Virginia Beach, Virginia; ISBN 0-9646747-1-8, 1995. *Available at http://m-selig.ae.illinois.edu/uiuc_lsat/Low-Speed-Airfoil-Data-V1.pdf*

**Comment RC2-48:**
"The rime- and glaze ice shapes investigated had a similar impact on the performance, both quantitatively and qualitatively. In the typical operating range, lift was reduced by 10% and drag increased by 80%." In which α range?

**The authors' reply to RC2-48:**
The typical α operation range for the S826 airfoil will be specified. It typically has the optimal L/D ratio at α = 7 deg.

**Comment RC2-49:**
"In the area before stall, lift..." by "In the regions before stall, lift..."

**The authors' reply to RC2-49:**
The suggested formulation is more precise and will be changed accordingly.

**Comment RC2-50:**
"In the area before stall, lift was reduced by 30% and drag was increased by 340%." specify α range.

**The authors' reply to RC2-50:**
The range suggested is from α = -2 to 5 degrees and will be specified.

**Comment RC2-51:**
"All ice types lead to performance losses of a magnitude that will reduce power output significantly." Please use quantitative conclusions. Calculate the effect on aerodynamic efficiency and power coefficient if you pretend to present conclusions on the influence in the rotor performance.

**The authors' reply to RC2-51:**
The authors agree that the statement should be rewritten with focus on changes in aerodynamic force coefficients rather than power output, as previously mentioned.

**Comment RC2-52:**
"The deviations are most pronounced after stall, leading to the assumption that they could be related to inaccuracies in the turbulence model and the absence of a transition model." Please support this conclusion on an evidence or on the work of other authors.

**The authors' reply to RC2-52:**
We agree with the reviewer that such a statement has to be supported with evidence or suitable references. As stated in the author's comment on **RC2-38**, the sim-exp deviations after stall probably cannot be blamed on the absence of a transition model anymore. To directly relate them to a "bad" turbulence model is also difficult, although the performance of RANS models for stalled airfoils is observed to be unsatisfactory (see e.g. Cakmakcioglu et al. (2014), who performed several types of simulations on the S826 airfoil). In Cakmakcioglu et al. (2014) it is shown that both RANS models (k-omega SST and the transitional Langtry-Menter gamma-Re-theta) do not perform well after the onset of stall. A Delayed Detached Eddy Simulation (DDES), however, gives good agreement with experimental results.
In conclusion, RANS models seem to have problems with the prediction of stalled airfoil flow in general. But it is a too vague statement to blame it on transition or turbulence models only.

Furthermore, three-dimensional flow effects in the experiment at the onset of stall could be a reason for a mismatch for exp. and sim. Results. Ref. Bartl et al. (2018)

References:

S. C. Cakmakcioglu, I. O. Sert, O. Tugluk, N. Sezer-Uzol, 2-d and 3-d cfd
620 investigation of nrel s826 airfoil at low reynolds numbers, J. Phys.: Conf.
Series 524 (2014) 012028. doi:10.1088/1742-6596/524/1/012028.

J. Bartl, K. Sagmo, T. Bracchi, L. Sætran. Performance of the NREL S826 airfoil at low to
moderate Reynolds numbers: A reference experiment for CFD models
Submitted to European Journal of Mechanics - B/Fluids (2018)

**Comment RC2-53:**
"...great potential in applying CFD icing methods..." CFD icing method is confusing, please
rephrase the sentence.

**The authors' reply to RC2-53:**
The authors agree and the sentence will be rephrased as follows: "...great potential in
applying
CFD methods to investigate icing and cold climate impacts on wind turbines."

**Comment RC2-54:**
"Further investigation on impact of ice extent, both in span and chord directions, would
provide useful insight to the total effect of icing on a wind turbine installation." Vague
conclusion. Please be more specific, what about the stochastic nature of the ice formation
process?

**The authors' reply to RC2-54:**
The authors agree and suggest the following reformulation: "Further investigation on impact
of ice extent, both in spanwise and chordwise directions, and consideration of the stochastic
nature of the ice formation process would provide extended knowledge on the total effect of
icing on a wind turbine installation."

**In general:**
In conclusions, we thank the reviewer for his/her comments. The reviewer's made
suggestions that, in the author's opinion, will improve the quality and scientific relevance of
the paper.